

# MieszkamTu
## MieszkamTu — platforma i forum miejskie dla samorządów



**Autorzy**: Mikołaj Jastrzębski ⓘ · Wiktor Stankiewicz ⓘ · Kacper Grobelny ⓘ · Damian Sawko ⓘ

**Opiekun:** dr inż. Bogumiła Hnatkowska

**Streszczenie**

Projekt MieszkamTu ma za zadanie zrewolucjonizować interakcję między mieszkańcami, a władzami miasta. Naszym celem jest umożliwienie każdemu obywatelowi wyrażania opinii i uczestnictwa w decyzjach dotyczących ich bezpośredniego otoczenia. Chcemy, aby władze miasta mogły szybko i transparentnie informować mieszkańców oraz lepiej rozumieć ich potrzeby.

Platforma rozwiązuje problem niskiego zaangażowania jednostek w procesy decyzyjne, co osłabia zaufanie do lokalnych władz. Dodatkowo ułatwia władzom zarządzanie opiniami mieszkańców, automatyzując analizę komentarzy i identyfikowanie kluczowych potrzeb społeczności.

Platforma składa się z takich modułów jak: Interaktywne mapy i wizualizacje, Inwestycje, Ogłoszenia, Nieruchomości, Automatyczna moderacja komentarzy i obliczanie ich sentymentu poprzez Sztuczną Inteligencję, Chatbot, Kreator Newslettera, Kalkulatory BAF (ang. Biotope Area Factor, czyli współczynnik obszaru biotopów) [6], wizualizacja zanieczyszczenia powietrza, modele 3D.

Projekt MieszkamTu usprawni efektywność pracy urzędników i poprawi jakość komunikacji z mieszkańcami, a użytkownikom zapewni szybki dostęp do informacji o wydarzeniach, inwestycjach oraz lepszą możliwość wyrażania opinii.

Założeniem projektu jest możliwość dotarcia aplikacji do wielu miast, prawna zgodność zarządzania oraz przechowywania danych.

# 1 ROZWINIĘCIE

## 1.1 Wstęp

Każdego dnia, gdy budzimy się i odkrywamy niezapowiedziany remont i zamknięcie drogi czy też zmianę trasy autobusu, odczuwamy frustrację. Ta jedna dziura w drodze, którą codziennie mijamy, nagle staje się symbolem niedziałającego systemu, w którym głos mieszkańców często pozostaje nieusłyszany.

Samorządy oraz rynek miejskich technologii stoi przed kluczowym problemem: brak zaangażowania mieszkańców w procesy decyzyjne. Statystyki CBOS z 2022 roku [1] rzucają światło na ten niepokojący obraz – 71% Polaków czuje, że nie ma wpływu na decyzje krajowe, a 50% na decyzje lokalne. To zjawisko obniża zaufanie do lokalnych władz i ogranicza poczucie wspólnoty. [3]

Problem ten jest globalny. Prognozy ONZ przewidują, że do 2050 roku 68% światowej populacji będzie mieszkać w miastach [7], co znacznie zwiększy potrzebę inteligentnych rozwiązań miejskich. Dodatkowo ponad 4,5 miliarda ludzi ma dostęp do internetu, z czego 75% oczekuje usług cyfrowych od władz [4], jednak tylko 10% uważa, że komunikacja z władzami miast jest skuteczna [5]. Jest to wyraźna szansa na wprowadzenie nowoczesnych platform zaangażowania obywatelskiego.

Prognozuje się, że rynek rozwiązań miejskich osiągnie wartość 2,57 bln USD do 2025 roku [9]. Ten fakt wskazuje na rosnące zapotrzebowanie na zaawansowane technologicznie usługi miejskie. Rządy na całym świecie, w odpowiedzi na potrzeby cyfryzacji usług publicznych i zwiększenia partycypacji obywatelskiej, coraz częściej inwestują w transformację cyfrową, w tym w platformy zaangażowania mieszkańców. Podsumowanie powyższych statystyk przedstawiono na Rys. 1 oraz Rys. 2

Polska, z ponad 2400 samorządami, oraz inne kraje na świecie, reprezentują olbrzymi potencjał dla platformy MieszkamTu. Zaprojektowana jako system modularny i elastyczny, nasza platforma może być łatwo dostosowana do różnorodnych potrzeb, czy też regulacji prawnych otwierając drzwi do międzynarodowej ekspansji.

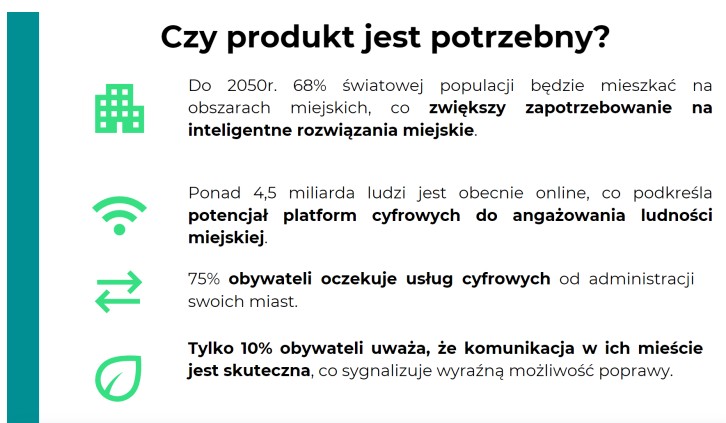

Rysunek 1: Omówienie statystyk dotyczących rynku

Rysunek 2: Omówienie, dlaczego projekt jest potrzebny

**Cele projektu i korzyści biznesowe**

- **Modularność i elastyczność**: Platforma MieszkamTu wyróżnia się na tle konkurencji dzięki modularnej i elastycznej architekturze, umożliwiając łatwą adaptację do specyficznych potrzeb każdego miasta, niezależnie od jego wielkości czy kraju.

- **Rozszerzenie świadomości społecznej i obywatelskiej**: MieszkamTu inicjuje globalny trend angażowania obywateli w sprawy miasta i kraju, dotychczas obserwowany głównie w krajach z demokracją bezpośrednią, jak Szwajcaria. Kraje te mają skonstruowane odpowiednie mechanizmy i systemy, aby zaangażować jak najszerszą grupę obywateli w proces decyzyjny na każdym poziomie (samorządu, kraju). MieszkamTu także udostępnia odpowiednie mechanizmy i systemy do przeprowadzania produktywnego dialogu. [8] Nasza platforma przyczyni się do rozszerzenia świadomości społecznej i obywatelskiej, tworząc lepsze warunki życia poprzez promowanie aktywnej partycypacji w życiu lokalnym.

- **Wszystko w jednym miejscu**: Zintegrowane środowisko dla wszystkich usług miejskich, zwiększające efektywność kosztową i operacyjną dla samorządów, a dla mieszkańców upraszczające dostęp do niezbędnych informacji i narzędzi.

- **Zwiększone zaangażowanie mieszkańców**: Możliwość udostępniania treści w serwisach społecznościowych i funkcjonalność komentowania na platformie zachęca do częstszego korzystania z forum, znacząco zwiększając zaangażowanie użytkowników w porównaniu do innych dostępnych rozwiązań.

## 1.2  Prace powiązane

Oferta konkurencji w zakresie narzędzi do zarządzania miastem i angażowania społeczności lokalnych jest zazwyczaj fragmentaryczna i nie oferuje tak wszechstronnego pakietu jak MieszkamTu. Alternatywne rozwiązania skupiają się na pojedynczych aspektach zarządzania miejskiego, takich jak zgłaszanie problemów infrastrukturalnych, czy komunikacja alarmowa, ale żadne z nich nie zapewnia tak szerokiego zakresu funkcji w jednej, zintegrowanej platformie.

Konkurencyjne rozwiązanie NaprawmyTo.pl koncentruje się na zgłaszaniu usterek i problemów lokalnych przez mieszkańców, ale nie umożliwia szerszej dyskusji na temat proponowanych inwestycji czy strategii rozwoju.

Aplikacja Blisko.co, chociaż jest popularna i ma znaczącą liczbę pobrań, również nie oferuje kompletnego pakietu usług komunikacyjnych, a jej funkcjonalność jest ograniczona do określonych dziedzin.

Portale informacyjne mogą dostarczać wiadomości na temat bieżących inwestycji, lecz brakuje im interaktywności oraz możliwości zaangażowania mieszkańców w procesy planistyczne i decyzyjne.

Ogłoszenia w urzędach, na stronach internetowych czy w Biuletynach Informacji Publicznej (BIP) oferują podstawową informację, lecz nie wspierają dwukierunkowej komunikacji ani nie budują społeczności wokół projektów miejskich. Więcej szczegółów na temat konkurencyjnych rozwiązań przedstawiono na Rys. 3 oraz Rys. 4

Projekt wyróżnia się na tle konkurencji kompleksowym pakietem usług, który łączy w sobie moduły inwestycji, ogłoszeń, nieruchomości, newsletterów oraz szczególnie unikalna funkcjonalność kalkulatora BAF, modeli 3D wraz z wizualizacją, usługę automatycznej moderacji i analizy sentymentu komentarzy oraz chatbotem. Jest to odpowiedź na potrzebę komunikacji, angażowania społeczności, zrównoważonego rozwoju oraz demokratycznego zarządzania w miastach.

Nasza platforma jest kompleksowym rozwiązaniem wspomagającym zarządzanie miastem, podczas gdy produkty konkurencyjne oferują jedynie wąskie fragmenty tego, co MieszkamTu dostarcza w jednym, spójnym rozwiązaniu.

## Konkurencja

**Facebook**
niezorganizowana dyskusja bez moderacji prowadzi do chaosu

**OTWOCK ALERT ONLINE**
Koncentruje się głównie na alertach i ogłoszeniach alarmowych, MieszkamTu zapewnia szerszy zakres usług w tej samej cenie.

**dRural**
Koncentruje się głównie na tworzeniu lokalnego rynku usług na obszarach wiejskich. Projekt rozpoczął się w 2021 r., ale nie ma rzeczywistych dowodów na to, że pomaga, MieszkamTu wystartowało w styczniu 2024 r. i będzie działać w Bieruniu już w maju.

**CITIFY.EU**
Dostępne tylko w Wilnie. Skupia się głównie na inwestorach. Nie bierze pod uwagę mieszkańców i dobrobytu miasta. Większość funkcji wiąże się z dużymi kosztami dla użytkowników. MieszkamTu jest darmowy dla wszystkich, ponieważ płaci za niego urząd miasta.

**MieszkamTu robi wszystko lepiej i taniej. Oferujemy znacznie więcej niż inni**

ponieważ...

Rysunek 3: Omówienie różnic między MieszkamTu a konkretnymi rozwiązaniami na rynku posortowane rosnąco wg istotności i podobieństwa

## Konkurencja - nasza odpowiedź

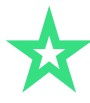

MieszkamTu wyróżnia się na tle konkurencji, oferując **kompleksowy pakiet usług**, zapewniając bezprecedensową **interaktywność i zaangażowanie** społeczności lokalnych. Platforma stanowi nie tylko narzędzie komunikacyjne, ale także platformę wspierającą **zrównoważony rozwój** i demokratyczne zarządzanie w miastach.

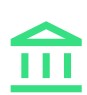

MieszkamTu jest **swobodnie dostępne** dla wszystkich obywateli, zapewniając, że platforma służy szerszej społeczności bez nakładania obciążeń finansowych na indywidualnych użytkowników. To inkluzywne, szeroko zakrojone podejście podkreśla zaangażowanie MieszkamTu w **poprawę życia w mieście dla wszystkich**, czyniąc go doskonałym wyborem dla **miast pragnących wspierać bardziej zaangażowaną, poinformowaną i spójną społeczność lokalną.**

Rysunek 4: Uzasadnienie wyższości MieszkamTu nad konkurencją

## 1.3 Rezultaty

**Osiągnięcia techniczne projektu**

- **Zaawansowana architektura systemu** - System został zaprojektowany w oparciu o trójwarstwową architekturę, składającą się z warstwy danych, logiki biznesowej oraz prezentacji.

- **Konteneryzacja systemu** - Poszczególne komponenty systemu zostały umieszczone w kontenerach technologii Docker oraz podzielone następująco:

  - **mitu-backend** - Warstwa logiki biznesowej zbudowana w NestJS przy użyciu TypeScript, składająca się z 19 modułów, serii interceptorów, dekoratorów oraz pipe'ów. W warstwie tej zaimplementowano integrację z zewnętrznymi serwisami Azure oraz połączenie z bazą danych z pomocą PrismaORM.
  - **mitu-frontend** - Warstwa prezentacji stworzona w React z wykorzystaniem TypeScript, zawiera dużą liczbę komponentów oraz serwisów stworzonych za pomocą technologii Zustand. Obejmuje także autorską modyfikację routera dla dynamicznej nawigacji oraz obsługi serii wydarzeń zależnych od miejsca w którym znajduje się użytkownik na stronie internetowej.
  - **mitu-postgres** - Kontener z bazą danych PostgreSQL, obsługujący przechowywanie danych aplikacji. Konteneryzacja bazy danych zapewnia izolację, co zwiększa bezpieczeństwo.
  - **mitu-cron** - Moduł napisany w technologii Spring (Java), odpowiedzialny za okresowe aktualizacje encji w bazie danych oraz usuwanie przestarzałych informacji.
  - **mitu-smtp4dev** - Serwis SMTP wykorzystywany w procesie rozwoju do obsługi systemu mailowego, odpowiedzialnego za rejestrację kont i newsletter.
  - **mitu-prerender** - Kontener umożliwiający wprowadzenie SEO (ang. Search Engine Optimization), koniecznego stronie typu Single Page Application. Poprawiają on indeksowanie przez wyszukiwarki.
  - **mitu-redis** - Kontener odpowiedzialny za caching zasobów na stronie, zwiększający wydajność aplikacji i zmniejszający obciążenie backendu.

- **Proces CI/CD (ang. Continuous integration, continuous deployment)** - Repozytorium projektu, znajdujące się na platformie Gitlab, zostało opatrzone kompleksowym procesem CI/CD. Zawiera on:

  - **Statyczny audyt jakości kodu**: Przed wdrożeniem każdej zmiany w publicznym repozytorium, kod testowany jest pod względem jakości i spójności z konwencją stylistyczną za pomocą narzędzi do analizy statycznej (Prettier, ESLint, Typescript)
  - **Testy automatyczne**: Przed wdrożeniem uruchamiane są wszystkie testy jednostkowe. Chroni to przed występowaniem błędów regresji.
  - **Automatyczna budowa kontenerów**: Wszystkie kontenery dockerowe budowane są każdorazowo przed wdrożeniem aplikacji.
  - **Automatyczne wdrożenie aplikacji**: Aplikacja wdrażana jest automatycznie na dwóch wirtualnych maszynach platformy Azure z pomocą skonfigurowanych runnerów platformy Gitlab.
  - **Automatyczne wdrożenie dokumentacji**: Wraz z wdrożeniem aplikacji, wdrażana jest również dokumentacja(link) serwera backendowego.
  - **Przechowywanie sekretów**: Platforma Gitlab w bezpieczny sposób przechowuje niejawne parametry aplikacji, takie jak klucze szyfrujące, klucze API czy hasło do bazy danych.

- **Baza Danych** - Baza danych projektu składa się z 15 tabel, obejmujących różne aspekty systemu, takie jak użytkownicy, posty, inwestycje, czy kategorie. Kluczową cechą jest wykorzystanie relacyjnych powiązań między modelami oraz wsparcie dla enumów, co zwiększa spójność i ułatwia walidację danych.

- **System Plików** - Autorski system plików umożliwiający zwięzłe przechowywanie i obsługę multimediów po stronie backend.

- **Autorski system uwierzytelniania** - Wdrożono dedykowany mechanizm uwierzytelniania korzystający z tokenów JWT (ang. JSON Web Token) celem uniezależnienia procesu autentykacji od zewnętrznych serwisów.

- **Integracja z mediami społecznościowymi** - Rozbudowano funkcjonalność projektu o możliwość udostępniania treści w mediach społecznościowych, co zwiększa zasięg i widoczność aplikacji.

- **Modele 3D inwestycji** - Wizualizacja inwestycji w formacie 3D pozwala użytkownikom lepiej wyobrazić sobie ich rzeczywisty wygląd i rozmieszczenie w przestrzeni.

- **Testy jednostkowe i interfejsu użytkownika** - Ze względu na złożoność projektu opracowano rozbudowany zestaw testów jednostkowych oraz testów UI (ang. User Interface), które poprawiają niezawodność systemu i wykrywają potencjalne błędy.

- **Obszerna dokumentacja** - Przygotowano szczegółową dokumentację backendu, udostępnioną jako czytelną stronę na platformie GitLab, z dokładnym opisem każdego pliku i modułu systemu.

- **Monitoring jakości powietrza** - Zintegrowano aplikację z miejskimi czujnikami jakości powietrza, umożliwiając użytkownikom sprawdzanie stanu powietrza w ich okolicy w czasie rzeczywistym.

- **Interaktywne mapy z punktami użyteczności publicznej i wizualizacje** - Mapy zawierają szczegółowe informacje o inwestycjach, ogłoszeniach i nieruchomościach w mieście. Wszystko to dopełnia możliwość wizualizacji inwestycji za pomocą modeli 3D.

- **Responsywność na urządzeniach mobilnych i komputerach** - Aplikacja została zaprojektowana w sposób zapewniający pełną responsywność, umożliwiając użytkownikom korzystanie z platformy na różnych urządzeniach, zarówno mobilnych, jak i stacjonarnych.

- **Modularność projektu** - System został opracowany w sposób umożliwiający jego szybkie wdrożenie w dowolnym mieście oraz dostosowanie do potrzeb mieszkańców i lokalnych urzędów, dzięki elastycznej i profesjonalnie zaprojektowanej architekturze.

- **Automatyczna moderacja komentarzy** - Integracja z usługą Azure Content Safety zapewnia skuteczną moderację treści, eliminując ryzyko pojawienia się nieodpowiednich tekstów czy obrazów w sekcjach komentarzy, co odciąża zarówno urzędników, jak i społeczność.

- **Analiza sentymentu** - Dzięki integracji z usługą Azure Text Analysis urzędnicy zyskują możliwość zrozumienia ogólnego nastawienia publicznego na podstawie komentarzy, bez konieczności przeglądania ich wszystkich, z jednoczesnym dostępem do najbardziej pozytywnych i negatywnych opinii.

- **Zaawansowany kreator newslettera** - Interaktywne narzędzie umożliwiające łatwe tworzenie, zapisywanie i wysyłanie newsletterów bogatych w różnorodne treści.

- **Graficzny kalkulator BAF** [2] - Pierwsze na rynku narzędzie umożliwiające ocenę wpływu budynku, projektu budowlanego czy też inwestycji na środowisko, szczególnie przydatne w podejmowaniu decyzji zrównoważonego rozwoju. Oferuje dwa tryby działania:

  - **Widok prosty** pozwala na szybkie wyliczenie współczynnika powierzchni biologicznie aktywnej na podstawie konkretnych wartości liczbowych.
  - **Widok graficzny** umożliwia intuicyjne obliczenie wskaźnika bez potrzeby znajomości szczegółowych wymiarów działki. Użytkownicy mogą w łatwy sposób zaznaczać różne elementy (powierzchnia domu, trawnika, drzewa, ścieżki) na mapie satelitarnej, co czyni narzędzie niezwykle przystępnym dla szerokiego grona użytkowników.

- **Chatbot** - Platforma posiada autorską implementację prostego chatbota, którego zadaniem jest udzielanie odpowiedzi na podstawowe pytania dotyczące punktów użyteczności publicznej znajdujących się w mieście.

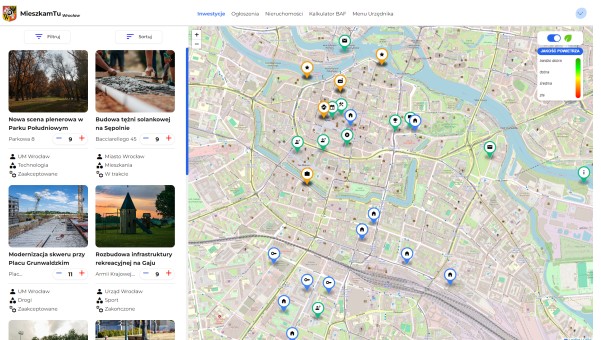

(a) Widok listy inwestycji

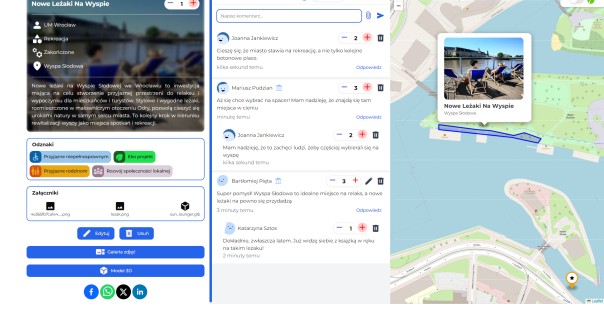

(b) Widok pojedynczej inwestycji z komentarzami

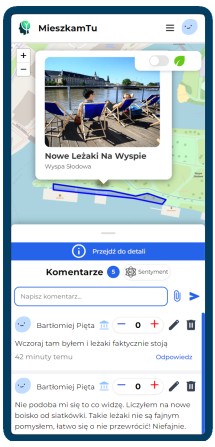

(a) Widok mobilny aplikacji

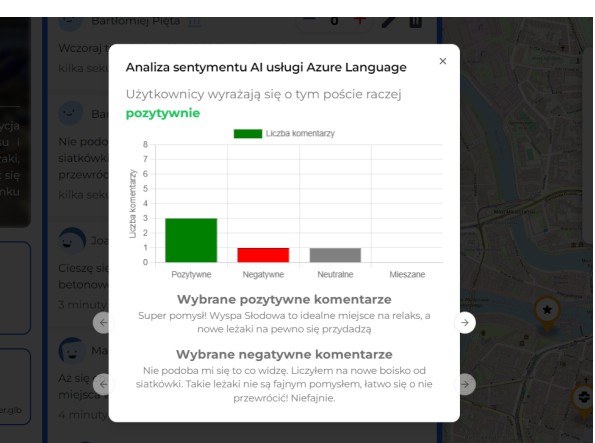

(b) Widok analizy sentymentu komentarzy

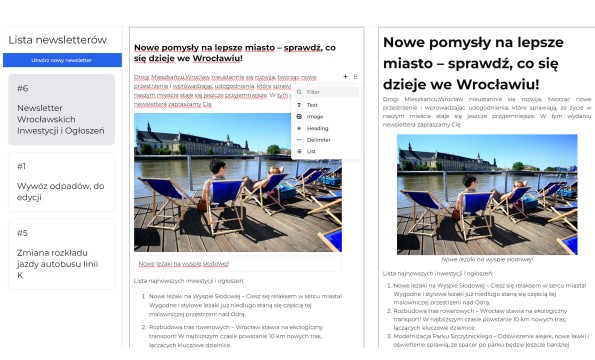

(a) Widok kreatora newslettera

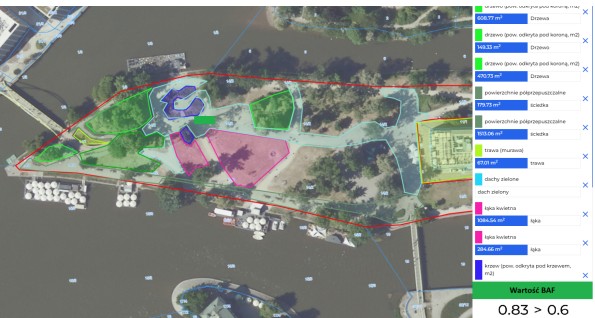

(b) Widok kalkulatora BAF w wersji graficznej

**Osiągnięcia biznesowe:**

- **Skalowalność i modularność:** Architektura systemu umożliwia dostosowanie funkcji do potrzeb samorządów różnej wielkości, co sprzyja potencjalnej ekspansji zarówno w kraju, jak i za granicą.

- **Wydajność i niezawodność:** System zapewnia stabilne działanie przy dużej liczbie jednoczesnych użytkowników (funkcjonował w mieście z 19334 mieszkańców), co zwiększa jego użyteczność w kontekście dużego miasta.

- **Zwiększenie zaangażowania mieszkańców:** Platforma umożliwia lepszy dostęp do informacji i narzędzi, znacząco podnosząc poziom interakcji między obywatelami a władzami samorządowymi.

**Korzyści dla użytkowników:**

- **Mieszkańcy:** Łatwy dostęp do informacji o inwestycjach, ogłoszeniach, nieruchomościach oraz stanie środowiska. Możliwość aktywnego uczestnictwa w procesach decyzyjnych za pomocą intuicyjnych narzędzi.

- **Urzędnicy:** Usprawnienie procesów moderacji, lepsze zarządzanie informacjami oraz szybsze i bardziej efektywne rozpoznawanie potrzeb mieszkańców dzięki analizie danych.

MieszkamTu rozwija się na bazie sukcesu wcześniejszej wersji projektu, która zdobyła uznanie na międzynarodowym hackathonie organizowanym przez PFR oraz Nordic Edge. Prototyp platformy został wdrożony w mieście Bieruń, gdzie znalazł praktyczne zastosowanie jako Interaktywna Mapa Bierunia (IMB).

# 2 ZAKOŃCZENIE

## 2.1 Wnioski

Sukcesy projektu MieszkamTu można pogrupować następująco:

- **Wszechstronność funkcji:** Platforma łączy różnorodne elementy, począwszy od informowania mieszkańców poprzez ogłoszenia, mapy interaktywne i modele 3D, aż po słuchanie opinii mieszkańców dzięki komentarzom, analizie sentymentu oraz moderacji treści. Jednocześnie zapewnia zaawansowane funkcje ekologiczne, takie jak kalkulatory BAF, w tym nowatorski graficzny kalkulator BAF, który umożliwia łatwe obliczenia bez potrzeby znajomości szczegółowych wymiarów działki.

- **Zrównoważony rozwój:** Projekt MieszkamTu w pełni wspiera cele zrównoważonego rozwoju, w szczególności cele ONZ [10] związane z miastami (Cel 11), innowacjami (Cel 9) i partnerstwami (Cel 17). Poprzez funkcjonalności wspierające ekologię, takie jak kalkulatory BAF, promowanie inwestycji przyjaznych środowisku itp. platforma staje się narzędziem do budowania świadomego i zrównoważonego rozwoju miast. Ponadto wspiera równy dostęp do dialogu społecznego dzięki przeniesieniu części dialogu nad kierunkiem rozwoju miasta do internetu.

- **Kompleksowość i innowacyjność:** Cała gama funkcji, zintegrowanych w jednym narzędziu, stanowi dowód na to, że MieszkamTu jest kompleksowym rozwiązaniem dla samorządów. Projekt znacząco usprawnia obecnie dostępne na rynku narzędzia komercyjne, oferując unikalne połączenie funkcjonalności w jednym systemie.

- **Praktyczne wdrożenie:** Wczesna wersja projektu została pomyślnie wdrożona w mieście Bieruń jako Interaktywna Mapa Bierunia (IMB), co stanowi dowód na praktyczną wartość i funkcjonalność rozwiązania.

MieszkamTu stanowi unikalne połączenie technologii, zrównoważonego podejścia oraz innowacyjnych rozwiązań, które mogą służyć zarówno mieszkańcom, jak i władzom miejskim, wyznaczając nowe standardy w zarządzaniu miastami. To niewątpliwie projekt o dużym potencjale, który nie tylko odpowiada na obecne potrzeby samorządów, ale także wyznacza nowe standardy w demokratyzacji procesów decyzyjnych

## 2.2 Przyszłe kierunki rozwoju

Przyszłe kierunki rozwoju MieszkamTu obejmują wiele możliwości, które mogłyby jeszcze bardziej zwiększyć wartość platformy:

**Nowe funkcjonalności:**

- Możliwość stworzenia modułu konsultacji społecznych online, umożliwiającego mieszkańcom udział w referendach, ankietach i publicznych dyskusjach. Aby zapewnić ich bezpieczeństwo oraz minimalizować ryzyko występowania fałszywych kont, integracja z dostawcami tożsamości cyfrowej, takimi jak mObywatel, byłaby potencjalnym krokiem.

- Możliwość wprowadzenia funkcji budżetu partycypacyjnego, który pozwalałyby mieszkańcom na bezpośrednie głosowanie nad alokacją części budżetu miasta.

- Rozbudowa wizualizacji danych z czujników miejskich, takich jak dynamiczne mapy jakości powietrza czy poziomu hałasu, dla lepszego zarządzania środowiskiem miejskim.

**Integracja zewnętrzna:**

- Istnieje możliwość integracji platformy z systemami administracji publicznej w celu automatyzacji procesów oraz zwiększenia efektywności zarządzania miastem.

- Integracja z innymi narzędziami miejskimi, jak np. aplikacje do monitorowania transportu publicznego, co stworzyłoby kompleksowy system zarządzania miejskiego.

**Dalsze innowacje:**

- Dodanie funkcji edukacyjnych, takich jak warsztaty online oparte na danych z platformy, wspierające świadomość społeczną i ekologiczną mieszkańców.

Przedstawione przez nas kierunki rozwoju pokazują potencjał MieszkamTu jako innowacyjnej platformy, która może nie tylko wspierać samorządy, ale również budować zaangażowanie i poprawiać jakość życia mieszkańców.

## 2.3 Podziękowania

Pragniemy podziękować wszystkim osobom zaangażowanym w rozwój projektu MieszkamTu.

- Dziękujemy opiekunowi projektu, Pani dr. inż. Bogumile Hnatkowskiej za wsparcie merytoryczne i pomoc w wymyśleniu nowych funkcjonalności projektu.

- Dziękujemy Pani dr. inż. Adriannie Kozierkiewicz za wsparcie merytoryczne i liczne praktyczne wskazówki.

- Dziękujemy Panu dr. inż. Marcinowi Pietrarnikowi za cenne wskazówki dotyczące tworzenia dokumentacji technicznej.

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
