# OpenReview forum: "MieszkamTu - Interaktywna Platforma i Forum Miejskie dla Samorządów"
_pwr.edu.pl/Wrocław_University_of_Science_and_Technology/2024/ZPI_Day — Wrocław University of Science and Technology 2024 ZPI Day Submission_

### Official Review · Reviewer_ps5M · 2024-12-04
**MieszkamTu**

**Confidence:** 5
**Significance Of Results:** 5
**Overall Quality:** 4

**Compliance With Template:**

4: High Quality – The article contains all the required sections, which are well-written and substantively correct, although minor errors or shortcomings may be present. The overall structure is clear and coherent.

**Description Of Results:**

5: Very High Quality – The results are described in detail, clearly and comprehensively, supported by thorough evaluation, analysis, and convincing usage examples. The description meets the highest substantive standards.

**Feedback On Consistency:**

Bardzo szeroka implementacja z wieloma wartymi uwagi innowacjami, jak analiza sentymentu komentarzy, automatyczna moderacja, interaktywne mapy. Zmieniłabym kolejność prezentacji niektórych punktów w listach punktowanych, aby oddzielić aspekty biznesowe od technicznych.

**Potential For Development:**

Projekt o zastosowaniach praktycznych, z dużym potencjałem wdrożenia, przygotowany do tworzenia nowych instancji dla kolejnych miast.

**Project Nature Evaluation:**

Projekt bardzo zaawansowany, zarówno od strony technologicznej, jak i funkcjonalnej. Z wdrożeniem - działa pod udostępnionymi przez studentów linkami w chmurze.

**Technical Language Precision:**

5: Very High Quality – The language is entirely appropriate for a technical report. All terms are used correctly and precisely, and the style is professional, clear, and coherent, without any errors or ambiguities.

---

### Official Review · Reviewer_kF6j · 2024-12-06
**Recenzja MieszkamTu**

**Confidence:** 4
**Significance Of Results:** 4
**Overall Quality:** 5

**Compliance With Template:**

4: High Quality – The article contains all the required sections, which are well-written and substantively correct, although minor errors or shortcomings may be present. The overall structure is clear and coherent.

**Description Of Results:**

3: Average Quality – The results are described with moderate detail. Some examples or evaluation elements are present but insufficiently developed or incomplete.

**Feedback On Consistency:**

Opis projektu jest spójny. Analiza problemu, cel i zakres funkcjonalny sposób został przedstawiony w zwięzły i jasny sposób. Prezentacja wyników jest przedstawiona w kontekście zdefiniowanych problemów. Podsumowanie i proponowana dalsza praca na projektem wydaje się być logicznym następstwem uzyskanych rezultatów.

**Potential For Development:**

Projekt ma bardzo wysoki potencjał rozwoju i komercjalizacji.

**Project Nature Evaluation:**

Jest to doskonały przykład inżynierii systemu informatycznego. Proces implementacji, oraz sama implementacja wydają się być na wysokim poziomie. Wybrane rozwiązania technicznie są wybrana świadomie, bez zbędnego nadmiaru i w oczywisty sposób realizują potrzeby projektu.

**Technical Language Precision:**

5: Very High Quality – The language is entirely appropriate for a technical report. All terms are used correctly and precisely, and the style is professional, clear, and coherent, without any errors or ambiguities.

---

### Official Review · Reviewer_C6YQ · 2024-12-06
**A multifunctional application with a great implementation potential**

**Confidence:** 4
**Significance Of Results:** 5
**Overall Quality:** 5

**Compliance With Template:**

5: Very High Quality – The article contains all the required sections, which are written in a very detailed, clear, and error-free manner. The structure is professional and meets expectations, and the content adheres to the highest substantive and formal standards.

**Description Of Results:**

5: Very High Quality – The results are described in detail, clearly and comprehensively, supported by thorough evaluation, analysis, and convincing usage examples. The description meets the highest substantive standards.

**Feedback On Consistency:**

The article presents a consistent and logical flow, where the identified problems are effectively addressed. However, a bit more clarification would be helpful regarding which features are specifically beneficial for residents, which ones for officials, and which serve both groups. Highlighting features that support communication between these groups and describing their benefits would also be a valuable addition to the paper.

**Potential For Development:**

The project demonstrates significant potential for development, especially as the first version of the application is already being used in a Polish city. The paper also highlights valuable suggestions for further development, indicating that the team has a strong understanding of the field and the practical needs of the project.

**Project Nature Evaluation:**

The project clearly exhibits characteristics of an engineering work, with a detailed description of solutions, and the selection of methods and technological solutions is both logical and well-justified. The project evaluation is also properly presented.

**Technical Language Precision:**

5: Very High Quality – The language is entirely appropriate for a technical report. All terms are used correctly and precisely, and the style is professional, clear, and coherent, without any errors or ambiguities.

---

### Decision · Program_Chairs · 2024-12-10

Accept (Poster)